# Perceived Supervisor Support for Health Affects Presenteeism: A Cross-Sectional Study

**DOI:** 10.3390/ijerph19074340

**Published:** 2022-04-04

**Authors:** Takahiro Mori, Tomohisa Nagata, Masako Nagata, Kiminori Odagami, Koji Mori

**Affiliations:** 1Department of Occupational Health Practice and Management, Institute of Industrial Ecological Sciences, University of Occupational and Environmental Health, Japan, 1-1 Iseigaoka Yahatanishi-ku, Kitakyushu 807-8555, Japan; tomohisa@med.uoeh-u.ac.jp (T.N.); m-nagata@med.uoeh-u.ac.jp (M.N.); k-odagami@med.uoeh-u.ac.jp (K.O.); kmori@med.uoeh-u.ac.jp (K.M.); 2Department of Occupational Medicine, School of Medicine, University of Occupational and Environmental Health, Japan, 1-1 Iseigaoka Yahatanishi-ku, Kitakyushu 807-8555, Japan

**Keywords:** perceived supervisor support for health, presenteeism, job demands-resources model, psychological state, leadership, health promotion program, health and productivity management

## Abstract

We investigated the relationship between perceived supervisor support for health (PSSH) and presenteeism by adjusting for psychological distress and employee work engagement. These are the mediators of the two paths shown in the job demands-resources model. A cross-sectional study was conducted using a questionnaire survey among 15,158 non-managerial employees from seven companies in Japan considered to have relatively high perceived organizational support for health (POSH). PSSH was evaluated with a single question, “My supervisor supports employees to work vigorously and live a healthy life”, on a four-point scale. Presenteeism was estimated using the quantity and quality method. Multilevel logistic regression analyses nested by company were conducted. Lower PSSH was more likely to be associated with presenteeism, but after adjusting for psychological distress evaluated by K6 and for work engagement, the relationship between PSSH and presenteeism weakened. Our results suggested that lower PSSH is linked to presenteeism through both psychological states because of its role as a resource, and other independent factors, even with relatively high POSH. Increased PSSH could act as a measure against presenteeism in the workplace. To achieve this, it is important to create an environment where supervisors can easily encourage employees to improve their health.

## 1. Introduction

Health-related productivity loss has become a major management and cost issue for companies, especially presenteeism. Health-related economic costs include direct costs such as medical costs and pharmaceutical costs, and indirect costs such as productivity loss. Absenteeism and presenteeism are often evaluated as workers’ health-related productivity losses. Absenteeism is “absence from work due to health problems” and presenteeism is defined as “health-related productivity loss while at paid work” [1]. Presenteeism has been shown to account for a large proportion of workers’ health-related costs. A survey of workers in companies in the United States estimated that the cost of presenteeism was 2.3 times the direct costs of medical care and drugs [2]. A survey among Japanese companies suggested that the cost of presenteeism was 64% of the total health costs [3]. Measures against presenteeism have, therefore, become an important issue for companies.

Previous studies on individual health problems that may cause presenteeism have covered both physical and mental health. Studies on physical health have investigated chronic diseases such as diabetes, high blood pressure, cancer, cardiovascular disease, inflammatory bowel disease, allergic disease, arthritis and migraine, and chronic symptoms such as neck and shoulder stiffness, lack of sleep, and dry eye. Other studies have examined mental illnesses such as depression and anxiety [2,3,4,5,6,7,8,9,10,11].

Psychosocial work environment factors affect the mental health of employees. Various occupational stress models have been proposed, one of which is the job demands-resources (JD-R) model [12]. This model proposes that job resources can reduce mental health problems and burnout by reducing job demands that cause physiological distress, or directly reducing the distress. Job resources also have a motivational role in improving employee work engagement (WE) [12,13,14,15,16]. Job resources have been reported to reduce presenteeism through these two psychological states of distress and WE [17,18].

Workplace resources that influence employee presenteeism include perceived organizational support (POS) and perceived supervisor support (PSS). POS is defined as global beliefs concerning the extent to which the organization values employee contributions and cares about their well-being [19]. PSS is defined as the extent to which employees believe their supervisors value their contributions, offer assistance, and care about their well-being [20]. POS and PSS are said to be correlated because supervisors act on behalf of the organization and employees become aware of the organization’s support through their supervisors’ actions [21,22]. However, Shi and Gordon investigated the impact of POS and PSS on employees’ WE and found that employees with high POS but low PSS had worse employee psychological contract breach and WE than those who showed low levels of both [23]. It is, therefore, suggested that POS and PSS need to be enhanced together, not just POS.

POS for health (POSH) and PSS for health (PSSH) are used specifically to indicate direct support for employee health. To date, studies have investigated the relationship between POSH, also known as perceived workplace health support (PWHS), and presenteeism. Employees with high POSH or PWHS have been reported to have lower presenteeism [24,25]. However, to our knowledge, no studies have investigated the relationship between PSSH and presenteeism.

In Japan, many companies are now actively supporting the health of their employees. The government has led the “health and productivity management (HPM)” initiative since 2014, including a “Certified HPM Corporation Recognition Program”. The certification program announces the top 500 companies that are actively promoting HPM [26]. The HPM certification process does not include employee perceptions, but it is expected that POSH will be relatively high in the top 500 companies, compared to others. We, therefore, investigated the relationship between PSSH and presenteeism for employees of companies among the top 500 companies with a relatively high POSH. PSSH is also considered to play a role as a resource, so we investigated the effect on presenteeism by adjusting the psychological distress and WE, which are the mediators of the two paths in the JD-R model.

## 2. Materials and Methods

### 2.1. Study Design and Participants

A cross-sectional study was conducted among workers from seven private companies, including four pharmaceutical companies, two in the manufacturing industry, and a service company in Japan. All companies were among the top 500 companies in the 2020 HPM Certification Program. The data were obtained from online questionnaire surveys conducted from 1 July 2020, to 31 December 2020. Of the 32,069 employees contacted, 19,695 responded. The purpose of this study was to evaluate perceived supervisor support, and thus the selected participants were 15,383 non-managerial employees. We excluded 225 participants who were missing data from the questionnaire survey, leaving data from 15,158 participants for analysis.

We explained the design and the purpose of this study to the employers and employees via e-mail or intranet homepage. We also explained that employees could choose whether to participate in this study or not, and obtained the informed consent of the participants. The questionnaire survey was conducted using a web questionnaire tool or a company-specific questionnaire form. The research protocol was approved by the Ethics Committee of Medical Research, University of Occupational and Environmental Health, Kitakyushu, Japan (H26-026).

### 2.2. Assessment of Perceived Supervisor Support for Health

Previous studies on POSH or PWHS evaluated one or two questions, and thus PSSH was evaluated with a single question drawing on previous studies [22,24,25], “My supervisor supports employees to work vigorously and live a healthy life”. Participants answered using a four-point scale: Strongly agree, Agree, Disagree, or Strongly disagree, and we categorized the responses as very high, high, low, and very low. We also carried out the analysis with PSSH as a continuous variable by defining very high PSSH as a score of 1 point, high as 2 points, low as 3 points, and very low as 4 points.

### 2.3. Assessment of Presenteeism

We used the quantity and quality (QQ) method to evaluate the productivity loss due to presenteeism [27]. In line with a previous study [28], the evaluation was made in several steps. First, we asked whether participants had experienced any health conditions while working during the past month. If the answer was “no”, presenteeism was set to zero. If the answer was “yes”, we asked the participants to identify their health problems from a list of 14 conditions and to select the condition that had most affected their work. If the conditions did not affect their work, presenteeism was also set to zero. The 14 conditions were as follows: (1) troubled by allergies (e.g., hay fever); (2) skin diseases/itchiness (e.g., eczema, atopic dermatitis); (3) disorders caused by infections (e.g., cold, influenza, gastroenteritis); (4) gastrointestinal disorders (e.g., recurrent diarrhea, constipation); (5) pain in arm and leg joints or lack of mobility (e.g., arthritis); (6) back pain; (7) painful neck or stiff shoulder; (8) headaches (e.g., migraine, chronic headache); (9) tooth trouble (e.g., toothache); (10) mental health problems (e.g., depression, anxiety); (11) insomnia, insufficient sleep; (12) a sense of weariness or fatigue; (13) eye problems (e.g., loss of vision, eyestrain, dry eye, glaucoma); and (14) other.

Second, we asked participants to describe the quantity and quality of their work when they were experiencing the health problem compared with when they had no problems. The answers were scored from 0 (unable to work at all) to 10 (normal). The presenteeism score was calculated using the following equation:Presenteeism score = 100 − quantity (range: 0–10) × quality (range: 0–10)

Drawing on a previous study [28], we defined the top 20% of responses as presenteeism in this study, which was, therefore, set as a presenteeism score of 44 or higher.

### 2.4. Assessment of Psychological Distress and Work Engagement

Psychological distress was measured using the Japanese version of the Kessler 6-Item Psychological Distress Scale (K6) [29]. Each item was measured on a five-point scale ranging from 0 (none of the time) to 4 (all of the time), with a minimum possible score of 0 and maximum possible score of 24.

Work engagement (WE) was measured using the nine-item Japanese version of the Utrecht Work Engagement Scale (UWES-9) [30]. Each item was measured on a seven-point scale ranging from 0 (never) to 6 (always/every day), with a minimum possible score of 0 and maximum possible score of 54.

### 2.5. Assessment of Covariates

Gender, age, and occupation were considered possible confounding factors. Age was expressed as a continuous variable. Occupation was classified into six categories: clerical; sales; research and development; engineering; production line; and other.

### 2.6. Statistical Analysis

Participant characteristics were summarized by PSSH category and by with and without presenteeism, using means and standard deviations (SDs) for continuous variables and percentages for categorical variables. We used unequal variance t-tests for age, K6 score, and WE score, and used Pearson’s Chi-square tests for gender and occupation to compare participant characteristics with and without presenteeism. We also created a boxplot of presenteeism score among PSSH category.

Multilevel logistic regression analyses were used to examine the relationship between PSSH and presenteeism. We set both PSSH as a continuous variable and PSSH as a categorical variable as independent variables, and presenteeism, that is, a presenteeism score of 44 or higher as the dependent variable. We estimated the odds ratios (ORs) using multilevel logistic regression analyses nested by company to assess the differences in POSH between companies. The ORs were estimated for the crude model (Model 1) and then adjusted for age, gender, and occupation (Model 2). We also adjusted first for K6 score (Model 3), and then both K6 and WE score (Model 4). A *p*-value of less than 0.05 (two-tailed) was considered statistically significant. All analyses used Stata Statistical Software (Release 16; StataCorp LLC, College Station, TX, USA).

We carried out a sensitivity analysis by performing the same analysis for a presenteeism score of 64 or higher, which was the top 10% of responses, and for a score of 30 or higher, which was the top 30% of responses, to check whether this gave similar results.

## 3. Results

Table 1 shows characteristics of the study participants by PSSH category. Of the 15,158 participants, 3125 (20.6%) reported very high PSSH, 8765 (57.8%) high PSSH, 2589 (17.1%) low PSSH, and 679 (4.5%) very low PSSH. The mean presenteeism score was the lowest (15.4) for very high PSSH and the highest (35.0) for very low PSSH, and the percentage of presenteeism (score of 44 or higher) was also the lowest (16.2%) for very high PSSH and the highest (41.2%) for very low PSSH. Presenteeism scores among PSSH category were shown in a boxplot (Figure 1).

Table 2 shows characteristics of the participants between with presenteeism and without presenteeism. Participants with presenteeism were younger, more women, higher in K6 score, and lower in WE score than participants without presenteeism.

Table 3 shows the relationship between PSSH as a continuous variable and presenteeism. There was a significant difference after adjusting for age, gender, and occupation (Model 2: OR = 1.56, 95% confidence interval (CI): 1.48–1.64, *p* < 0.001). The OR decreased, but there was still a significant difference after additional adjustment for K6 score (Model 3: OR = 1.25, 95% CI: 1.18–1.32, *p* < 0.001), and then WE score (Model 4: OR = 1.14, 95% CI: 1.08–1.21, *p* < 0.001).

Table 4 shows the relationship between PSSH as a categorical variable and presenteeism. The ORs for high, low, and very low PSSH were significantly higher than very high in Model 2. The ORs decreased, but were still all significantly higher in Model 3. In Model 4, there was no significant difference for high PSSH, but there was a significant difference for low (OR = 1.34, 95% CI: 1.16–1.55, *p* < 0.001) and very low (OR = 1.36, 95% CI: 1.10–1.67, *p* = 0.004).

In the sensitivity analysis, in the evaluation of the top 10% of presenteeism scores (a score of 64 or higher), there were significant differences in Model 2 and in Model 3 with PSSH as a continuous variable (Appendix A). In the analysis with PSSH as a categorical variable, the ORs in all categories was significantly higher than very high PSSH in Model 2 and the ORs for low and very low PSSH were significantly higher in Model 3. However, there were no significant differences in Model 4 (Appendix A). In contrast, in the evaluation of the top 30% of presenteeism scores (a score of 30 or higher), there was a significant difference in the analysis with PSSH as a continuous variable (Appendix A) and the ORs in all categories were significantly higher in the analysis with PSSH as a categorical variable, even in Model 4 (Appendix A).

## 4. Discussion

Our results showed that lower PSSH is more likely to be associated with presenteeism in companies that are actively engaged in health promotion activities, that is, companies that are considered to have relatively high POSH. After adjusting for psychological state of distress evaluated by K6, and for WE, the mediators of the two paths in the JD-R model, the relationship between PSSH and presenteeism weakened. This result suggested that PSSH may influence the decrease in presenteeism through psychological states because of its role as a resource. However, there was a significant difference for low and very low PSSH even after adjusting for K6 and WE, suggesting that factors other than psychological states affect presenteeism. A similar tendency was seen in the sensitivity analysis. When the top 10% of the presenteeism scores were used, no significant difference was observed. However, a tendency was observed in the analysis with PSSH as a continuous variable (OR = 1.07, *p* = 0.056), and when the top 30% were used, a significant difference was observed.

Our results suggest that PSSH has a role as a resource under the JD-R model, and may influence presenteeism through psychological state. Laing et al. showed that the relationship between POSH and presenteeism was mediated by mental health problems such as anxiety and depression [25]. In our study, PSSH showed similar results. Studies have reported that PSS reduces emotional exhaustion and burnout because of employee stress and presenteeism [31,32], and enhances WE and job satisfaction [33,34,35]. Jimenez et al. suggested that health-promoting leadership contributes strongly and directly to employees’ resources [36]. It indirectly affects the reduction of stress and burnout through adjustment of work environment and conditions, for example, by supervisors encouraging employees to take a day off, managing working hours, and considering the amount and content of work to support the health of employees [36]. Employees’ perception of this support for their health from their supervisor can lead to improved trust in their supervisor and better communication, which can lead to reduced psychological stress and increased WE. As a result, presenteeism may be reduced.

Our results suggested that PSSH is associated with presenteeism independently of psychological state of distress and WE. One possible link may be that health support from supervisors can maintain and improve the physical health of employees, which has a positive effect on work productivity. Workplace health promotion programs lead to clinically meaningful behavior modification of employees, such as promotion of healthy behavior and reduction of risk factors. This leads to reduced presenteeism [37]. The effectiveness of such programs is greatly affected by employee participation, but there are significant differences between workplaces and situations [38]. In particular, PSS has an important influence on employee participation in these programs: there are many reports that higher PSS is associated with higher participation rates [39,40,41,42]. PSSH, including explaining the purpose of company-provided health programs and encouraging employee participation, can increase the participation rate. This may affect the physical health of employees and possibly reduce presenteeism.

In addition to being involved in health promotion programs, supervisors may informally check up on employees’ health and physical condition each day. If there is a change, they may recommend that they take a day off, or encourage them to see a doctor. Early medical examination and treatment can often prevent diseases from becoming more severe, and may, therefore, reduce presenteeism. 

Our results show the importance of increasing PSSH as a measure against presenteeism in the workplace. Smidt et al. reported that employees’ participation in workplace wellness programs was lower when employees had high POS but low PSS than when both POS and PSS were low [43]. It is possible that the awareness and behavior of supervisors vary even in companies that are actively engaged in workplace health promotion activities such as the companies in this study. It is, therefore, necessary to develop various health measures and target improvements in both POSH and PSSH through changing the behavior of supervisors.

To increase PSSH, it is important to increase the POSH of supervisors first. Shanock et al. reported that the POS of supervisors had a positive relationship with the PSS of other employees, and that the PSS of other employees had a positive relationship with their POS, and both in-role and extra-role performance [44]. Edmunds et al. emphasized that lack of interest in or not knowing the program for line managers affects employees’ non-participation in physical exercise programs [40]. It is, therefore, important for supervisors to understand their company’s HPM, and to participate in health promotion programs and experience health support from the company.

It is also important to give supervisors a leadership role in workplace health promotion programs. Organizations that hold managers and supervisors responsible for the program have a higher percentage of employee program satisfaction and POS than organizations that do not [45]. Justesen et al. made suggestions for supervisors to play the role of workplace leaders [41]. It is also necessary to create an environment in which supervisors can easily encourage other employees to improve their health.

This study had several limitations. First, the participants were employees of seven large companies in Japan; thus, generalization may be limited. However, in recent years, the number of small- and medium-sized companies engaged in health management has increased considerably [46], and our results may, therefore, be more widely useful. They will also be useful for companies working on HPM in the future. Second, there may be some possible biases in this study. The overall response of this study was 19,695 out of 32,069 employees (61.4%), suggesting that, for example, those with poor physical condition could not participate. In addition, it is possible that there was a reporting bias, especially in the evaluation of PSSH. However, we explained to participants that the survey was anonymized and not personally identifiable, so we expect the impact to be small. Third, PSSH was evaluated with a single question and the validity of the measure was untested against the original concept of PSSH. There is no established index to measure PSSH; therefore, we drew on previous studies of POSH and PWHS [22,24,25]. Further research is needed to rigorously develop and validate the measure for PSSH. Fourth, because this was a cross-sectional study, we could not evaluate the causal relationship between PSSH and presenteeism. Employees’ feelings that their supervisors are supportive are also related to their health satisfaction [39]. PSSH may increase when both health satisfaction and productivity are high. It is, therefore, necessary to further investigate the causal relationship between PSSH and presenteeism. Fifth, the results for each index may also vary depending on employees’ physical and mental condition and their relationship with their supervisors at the time of the survey. In particular, since our survey was conducted under the COVID-19 pandemic, it is possible that the results were slightly different from the survey under normal circumstances.

## 5. Conclusions

Our study suggests that even with relatively high POSH, lower PSSH is associated with presenteeism, perhaps through psychological state and other independent factors such as physical health. It is, therefore, important to increase both POSH and PSSH as a measure against presenteeism and in companies that promote HPM. This requires companies to create an environment in the workplace in which supervisors can easily encourage other employees to improve their health.

## Figures and Tables

**Figure 1 ijerph-19-04340-f001:**
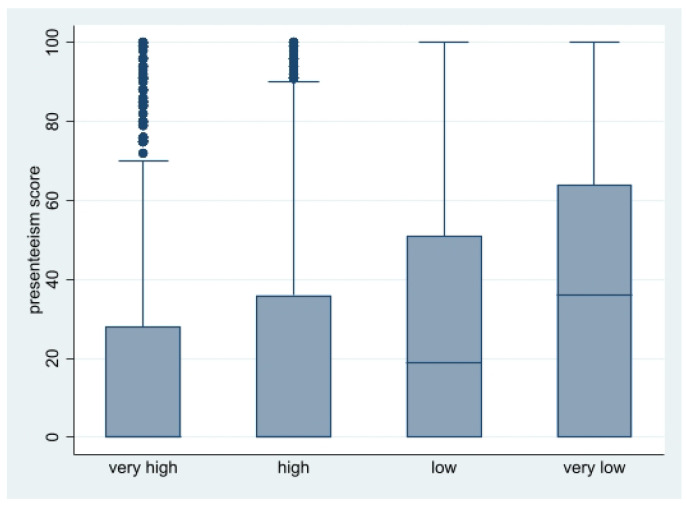
Boxplot of presenteeism score among PSSH category.

**Table 1 ijerph-19-04340-t001:** Characteristics of the study participants by categories of perceived supervisor support for health.

	Perceived Supervisor Support for Health			
	Very High	High	Low	Very Low
Number of participants	3125	8765	2589	679
Age, mean (SD)	41.9 (11.1)	43.4 (10.7)	44.3 (10.3)	43.5 (9.8)
Gender, men	2321 (74.3%)	6340 (72.3%)	1821 (70.3%)	502 (73.9%)
Occupation				
Clerical	792 (25.3%)	2189 (25.0%)	665 (25.7%)	159 (23.4%)
Sales	1243 (39.8%)	2384 (27.2%)	504 (19.5%)	102 (15.0%)
Research & Development	598 (19.1%)	1919 (21.9%)	523 (20.2%)	148 (21.8%)
Engineering	214 (6.8%)	1000 (11.4%)	381 (14.7%)	95 (14.0%)
Production line	145 (4.6%)	724 (8.3%)	309 (11.9%)	105 (15.5%)
Other	133 (4.3%)	549 (6.3%)	207 (8.0%)	70 (10.3%)
K6 score (range: 0–24), mean (SD)	3.0 (3.9)	4.2 (4.3)	6.0 (5.1)	8.7 (6.4)
Work engagement score (UWES-9) (range: 0–54), mean (SD)	33.6 (9.2)	27.2 (7.6)	22.7 (8.2)	18.3 (10.2)
Quantity of work (range: 0–10), mean (SD)	9.0 (1.8)	8.8 (1.9)	8.3 (2.1)	7.7 (2.6)
Quality of work (range: 0–10), mean (SD)	9.0 (1.8)	8.8 (1.9)	8.2 (2.2)	7.8 (2.5)
Presenteeism score, mean (SD)	15.4 (25.5)	19.6 (27.2)	27.4 (30.0)	35.0 (33.0)
Presenteeism (score of 44 or higher)	506 (16.2%)	1890 (21.6%)	826 (31.9%)	280 (41.2%)

UWES: Utrecht Work Engagement Scale; SD: standard deviation. Presenteeism score = 100 − Quantity of work × Quality of work.

**Table 2 ijerph-19-04340-t002:** Characteristics of the participants between with presenteeism and without presenteeism.

	With Presenteeism	Without Presenteeism	*p*-Value
Number of participants	3502	11,656	
Age, mean (SD)	42.5 (10.3)	43.5 (10.9)	<0.001
Gender, men	2357 (67.3%)	8627 (74.0%)	<0.001
Occupation			0.001
Clerical	913 (26.1%)	2892 (24.8%)	
Sales	900 (25.7%)	3333 (28.6%)	
Research & Development	728 (20.8%)	2460 (21.1%)	
Engineering	431 (12.3%)	1259 (10.8%)	
Production line	283 (8.1%)	1000 (8.6%)	
Other	247 (7.1%)	712 (6.1%)	
K6 score (range: 0–24), mean (SD)	7.0 (5.2)	3.7 (4.2)	<0.001
Work engagement score (UWES-9) (range: 0–54), mean (SD)	24.1 (8.8)	28.3 (8.9)	<0.001

UWES: Utrecht Work Engagement Scale; SD: standard deviation. Presenteeism score = 100 − Quantity of work × Quality of work.

**Table 3 ijerph-19-04340-t003:** Relationship between perceived supervisor support for health as a continuous variable and presenteeism.

	Model 1			Model 2			Model 3			Model 4		
	OR	95% CI	*p*-Value	aOR	95% CI	*p*-Value	aOR	95% CI	*p*-Value	aOR	95% CI	*p*-Value
PSSH (continuous)	1.54	1.46–1.62	<0.001	1.56	1.48–1.64	<0.001	1.25	1.18–1.32	<0.001	1.14	1.08–1.21	<0.001
K6 score (continuous)							1.14	1.13–1.15	<0.001	1.13	1.12–1.14	<0.001
WE score (continuous)										0.97	0.97–0.98	<0.001

Model 1: crude model. Model 2: adjusted for age, gender, and occupation. Model 3: Model 2, additionally adjusted for K6 score. Model 4: Model 3, additionally adjusted for work engagement score. All analyses used multilevel logistic regression nested by company. PSSH: perceived supervisor support for health; WE: work engagement; aOR: adjusted odds ratio; CI: confidence interval.

**Table 4 ijerph-19-04340-t004:** Relationship between perceived supervisor support for health as a categorical variable and presenteeism.

	Model 1			Model 2			Model 3			Model 4		
	OR	95% CI	*p*-Value	aOR	95% CI	*p*-Value	aOR	95% CI	*p*-Value	aOR	95% CI	*p*-Value
PSSH (categorical)												
Very high	Ref			Ref			Ref			Ref		
High	1.39	1.24–1.55	<0.001	1.41	1.26–1.57	<0.001	1.22	1.09–1.36	0.001	1.08	0.96–1.21	0.206
Low	2.31	2.03–2.62	<0.001	2.36	2.07–2.69	<0.001	1.64	1.43–1.88	<0.001	1.34	1.16–1.55	<0.001
Very low	3.46	2.88–4.15	<0.001	3.56	2.97–4.28	<0.001	1.78	1.46–2.18	<0.001	1.36	1.10–1.67	0.004
K6 score (continuous)							1.14	1.13–1.15	<0.001	1.13	1.12–1.14	<0.001
WE score (continuous)										0.97	0.97–0.98	<0.001

Model 1: crude model. Model 2: adjusted for age, gender, and occupation. Model 3: Model 2, additionally adjusted for K6 score. Model 4: Model 3, additionally adjusted for work engagement score. All analyses used multilevel logistic regression nested by company. PSSH: perceived supervisor support for health; WE: work engagement; aOR: adjusted odds ratio; CI: confidence interval.

## Data Availability

The data presented in this study are available on request from the corresponding author. The data are not publicly available because of legal and privacy issues.

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
