# Peer review of "Perceived Supervisor Support for Health Affects Presenteeism: A Cross-Sectional Study"

_ijerph, 2022, doi:10.3390/ijerph19074340_

Round 1
Reviewer 1 Report
An interesting paper presented in a succinct manner.
Author Response
Thank you very much for reviewing and evaluating.
Reviewer 2 Report
Dear Authors,
here below are some suggestions to improve the readability of the manuscript.
In Method section, what about the consent form? How did you manage possible discomfort experienced by participants? Have you suggested going to counseling centers, for example? In this type of investigation, participants could experience a negative feeling, distress and so on, thus it’s essential to offer support for those who need help.
In the Limits section, I suggest inserting the possible biases in this investigation: for example, the bias in participation, the social desirability that could influence reply to items.
Moreover, the investigation was conducted in the pandemic period. Any consideration about the possible influence in participation, reply to items, perceived support?
Reviewer 3 Report
The study used a sample of 15,158 non-managerial employees from seven companies in Japan. The association between perceived supervisor support for health (PSSH) and presenteeism was estimated using the Quantity and Quality method. The study used presenteeism>44 as the dependent variable and PSSH as the main independent variable to run non-adjusted and adjusted models.
Overall, this is a fascinating study and valuable for all organizations after pandemic. Here are a few observations:
Abstract: Before talking about the OR, describe your method and report the results in the abstract. Please write a few last sentences non-technically so the general audience can understand the meaning of this study. Please add the study's location in the abstract.
Introduction: the authors developed it well. I have no comments.
Method: I have a few points in the method section. Here are my comments.
- The author mentioned that they had used email or the internet to collect information. I understand the email, but please explain how data has been collected through the internet, any specific website, etc.
- The author used the logistic regression to estimate the association between presenteeism and PSSH, however, the method needs more clarifications; please describe the dependent variable and model e.g., presenteeism=f(BSSH, Control variables).
- Alternatively, or as a sensitivity analysis, I would use a negative binomial regression model (NBRG) using the presenteeism and report the IRR for PSSH.
Result section: This section needs some modifications, here are some comments:
- Add the categorical presenteeism to T1, to show % of the population with presenteeism>44.
- I would add a new descriptive table based on the presenteeism category (>44 and <44), repeat the T1, and then run an un-equal ttest to compare two populations.
- Table 2, is so confusing; the way I am reading this table, the authors have included the PSSH as a continuous and categorical variable, I do expect to see two models, one with continuous and one with categorical; please explain.
- I would add a boxplot to show the presenteeism score among PSSH groups and categorical presenteeism.
Discussion: There are no more comments, but you may need to modify this section based on the comments mentioned above.
Round 2
Reviewer 3 Report
Thank you very much for addressing my comments very well, good job!